# Effect of Sex, Age and Temperature on the Functional Response of *Macrolophus pygmaeus* Ramber and *Nesidiocoris tenuis* Reuter (Heteroptera: Miridae) on Eggs of *Tuta absoluta*

**DOI:** 10.3390/insects15070485

**Published:** 2024-06-28

**Authors:** Eleni Yiacoumi, Nikos A. Kouloussis, Dimitrios S. Koveos

**Affiliations:** Laboratory of Applied Zoology and Parasitology, School of Agriculture, Aristotle University of Thessaloniki, 54124 Thessaloniki, Greece; egiakoumi@agro.auth.gr (E.Y.); nikoul@agro.auth.gr (N.A.K.)

**Keywords:** *Macrolophus pygmaeus*, *Nesidiocoris tenuis*, Miridae, *Tuta absoluta*, functional response, tomato, sex, adult age, temperature, predator

## Abstract

**Simple Summary:**

The predatory insects *Macrolophus pygmaeus* Ramber (Heteroptera: Miridae) and *Nesidiocoris tenuis* Reuter (Heteroptera: Miridae) are important biological control agents used commercially for the control of main pests in greenhouses, such as *Tuta absoluta* Meyrick (Lepidoptera: Gelechiidae) and whiteflies (Homoptera: Aleyrodidae). In this study, we investigated the predation efficacy of young and old females and males of the two mirid bugs, *M. pygmaeus* and *N. tenuis,* when feeding on eggs of *T. absoluta* on tomato leaves. Young females of *M. pygmaeus* exhibited a higher predation efficiency compared to old ones, whereas males displayed a consistently low efficiency of predation irrespective of their age. Both young females and males of *N. tenuis* displayed a similarly high predation efficiency, although the old females exhibited a higher efficiency of predation compared to their male counterparts, but lower than the efficacy of the *M. pygmaeus* individuals. Our findings indicate that the two predatory species have different functional response characteristics to their prey depending mainly on their sex and age which may influence their efficacy as biological control agents.

**Abstract:**

The predatory mirids *Macrolophus pygmaeus* Ramber (Heteroptera: Miridae) and *Nesidiocoris tenuis* Reuter (Heteroptera: Miridae) are used for the biological control of *Tuta absoluta* Meyrick (Lepidoptera: Gelechiidae) and other pests in tomato greenhouses. The functional response of 1-day-old (young) and 10-day-old (old) adult females and males of *M. pygmaeus* and *N. tenuis* on eggs of *T. absoluta* was determined on tomato at two temperatures (20 °C and 25 °C) and LD 16:8. Females of *M. pygmaeus* exhibited higher predation efficiency than males at both tested temperatures. Young *M. pygmaeus* females had a higher efficiency than old ones, whereas males had a low efficiency irrespective of age. The predation efficiency of *N. tenuis* was high (but lower than *M. pygmaeus*) in both young females and males, although old females had a higher efficiency than the respective males. Our results show that the two predatory species have different functional response characteristics to their prey depending mainly on sex and age, which may affect their role as biological control agents.

## 1. Introduction

The dicyphine mirid bugs *Macrolophus pygmaeus* Ramber and *Nesidiocoris tenuis* Reuter (Heteroptera: Miridae) are native to the Mediterranean basin and commercialized as beneficial insects in biological pest control systems [1,2,3]. The mirid *M. pygmaeus* has been utilized for nearly three decades, predominantly in augmentative biological control in greenhouses, to regulate small vegetable crop pests such as spider mites, whiteflies, aphids, thrips, and lepidopteran pests (mainly eggs and young larvae) [2,3,4]. Despite being identified as an actor in biological control later, *N. tenuis* has gained significant attention due to its great predatory capacity and performance at high temperatures [5,6]. The capacity of the two species as biocontrol agents against *Tuta absoluta* Meyrick (Lepidoptera: Gelechiidae) has been investigated and highlighted [7,8]. Both *M. pygmaeus* and *N. tenuis* exhibit zoophytophagy, a special type of omnivory, with the ability to feed on both plant tissue and a broad spectrum of arthropods [9]. The presence of both peptidase and amylase in the salivary glands has been associated with this polyphagous feeding capacity [10]. These two species have a proclivity for zoophagy, although they facultatively exhibit phytophagy, which may play a substantial role in their preservation and fitness [9,11,12,13,14,15,16,17,18,19]. The ability to utilize both trophic sources offers a significant advantage when it comes to the desired early establishment of the predator in the crop before pest infestation proliferates [9]. The mirid, *M. pygmaeus*, has the added ability to complete its life cycle without access to prey, feeding exclusively on the host plant tissue of tomato and a multitude of cultivated vegetable plants [9]. In contrast, there are few recorded plant species, notably those native to tropical and subtropical regions, *Sesasum indicum* Linnaeus (Pedaliaceae) and *Cleome hassleriana* Chod. (Cleomaceae) that are suitable banker plants for the preyless development and oviposition of *N. tenuis* [20,21]. Interestingly, *N. tenuis* is considered to possess a greater voracity than *M. pygmaeus*, and in cases of high population densities, may cause economically important plant damage and yield loss in the form of aborted flowers and fruit and necrotic rings due to phytophagy on vegetative and reproductive plant parts [22,23,24,25]. As such, the status of *N. tenuis* as a pest or beneficial insect has been a point of contention [26,27]. However, in the Mediterranean region, the benefits emanating from its high generalist predation capacity, especially on more than one tomato pest, such as the invasive pest *T. absoluta* and *Trialeurodes vaporariorum* Westwood (Homoptera: Aleyrodidae), have been argued to outweigh the danger of plant feeding-associated damage [9]. 

The micro lepidopteran pest, *T. absoluta*, is considered one of the most devastating pests of tomato, along with *T. vaporariorum*. Native to South America, it has since extended its presence to Europe (beginning with Spain in 2006), Africa (starting with North Africa, Tunisia, and Morocco in 2008), and Asia (starting with Turkey in 2009) [28,29]. The larvae are voracious and feed on all aerial plant parts in protected galleries within leaves, stems, and fruit. When appropriate control measures are not implemented, infestation can lead to complete crop failure [29]. Certain characteristics associated with the ecology and the biology of this pest also favour the development of resistance to a broad range of insecticides, rendering control ineffective [30,31,32,33,34]. The inadequacy of chemical control in delivering sufficient crop protection serves as a catalyst for endeavours aimed at developing and implementing alternative control methods, including the utilization of beneficial insects, as integral components of Integrated Pest Management (IPM) [35].

Functional response experiments have been a longstanding tool for evaluating the effectiveness of biocontrol agents in controlling arthropod pests and defining the range of pest densities and conditions within which the biological control agent remains effective. The functional response of a predator to prey density describes the per capita feeding rate of a predator as a function of resource density and is classified according to the three types described by Holling, 1959 [36]. Both Type II and Type III functional responses of predators are desired characteristics for beneficial arthropods utilized as biological control agents and have been reported for many mirid predators, with the Type III functional response being defined by a learning behaviour [37,38,39,40,41]. Two parameters, the attack rate coefficient (α) (ability to capture) and the handling time (T*_h_*) (ability to handle and consume prey), are used to provide insight into the functional response of the predator. In particular, in the case of females, such foraging performance information is impactful, as it directly controls both the quantity and quality of offspring, thus influencing the establishment of predators, the dynamics of the populations, and the overall predation efficacy [42]. 

Various functional response experiments have been conducted to elucidate the efficacy of the two species when consuming a multitude of small arthropods, with the results depending on factors such as host plant, spatial complexity, prey item, prey instar, and temperature [39,40,41,43,44,45,46,47,48,49]. 

The primary objective of this study was to assess the functional response parameters of the two species, *M. pygmaeus* and *N. tenuis*, against *T. absoluta* targeting the egg stage and to provide new knowledge about their predation efficacy based on this. Functional response experiments were conducted at six predefined densities of the synchronous prey *T. absoluta* eggs under laboratory conditions, considering the impact of adult sex and predator ageing on predation, at two experimental temperatures. 

## 2. Materials and Methods

### 2.1. Insect Colonies

The stock colonies of *M. pygmaeus* and *N. tenuis* were initiated from individuals collected from rural Thessaloniki and Crete, respectively, and kept for 10 generations in the laboratory before their use in the experiments. The colonies were maintained in the laboratory in cylindrical plastic cages (diameter, 25 cm, height, 25 cm), with fresh bean pods as oviposition substrate and dehydrated *E. kuehniella* Zeller (Lepidoptera: Pyralidae) eggs ad libitum at 25 °C and a photoperiod of 16:8 (L:D). 

The stock colony of *T. absoluta* was established with adults that emerged from heavily infested tomato leaves collected from a commercial plantation in northern Greece. The colony was maintained on tomato plants in tent cages at 25 °C and a photoperiod of 16:8 (L:D).

Predation experiments were conducted at two temperatures (20 °C and 25 °C), a photoperiod of 16:8 (L:D), and 65 ± 5% RH; these conditions were favourable for the development and reproduction of the two predatory species. One- or 10-day-old adult females and males of either *M. pygmaeus* or *N. tenuis* from the laboratory colonies were kept in an empty rearing cage, with access only to water for 24 h, before their use in the functional response experiments.

### 2.2. Bioassay

For the functional response experiments, tomato leaflets of comparable size (approximately 6 cm in height and 3 cm in width) were excised from tomato plants, and their petioles were enveloped in dampened cotton wool in plastic vials (5 cm in height; 3 cm in diameter) with water. The vials with the leaflets were maintained for 1 to 3 h in an ovipositional cage (60 cm × 60 cm × 60 cm) with approximately 300 *T. absoluta* adults from the laboratory colony for egg laying. Subsequently, the number of eggs laid on both the upper and underside of each leaflet was scored and delimited to reach the desired density. The complete leaflet served as an experimental arena. In functional response experiments, the predator’s mobility is limited to the controlled experimental arena, whereas in natural agroecosystems, both predator and moving prey stages roam freely. Our arenas were adopted in an attempt to align the experimental conditions more closely with field conditions, where the predator consumes *T. absoluta* eggs deposited on both the upper and underside leaf surfaces. The egg densities were selected through preliminary tests to ensure maximum levels of predation. The selected egg densities for the functional response experiments were 20, 40, 60, 80, 100, and 120 eggs.

The tomato leaflets with the eggs were transferred into individual cages consisting of inverted plastic cups (volume, 500 cm^3^) with dual openings on opposite sides of their cylindrical walls, covered with fine mesh, to facilitate air circulation. Each cup was sealed with a bottom lid and included a circular aperture at the top, allowing the introduction of the predator individual into the experimental arena before being sealed with a cork. 

The experimental setup comprised an individual cage with one leaflet positioned in the vial, with a specific egg density, to which either a previously starved male or female of *M. pygmaeus* or *N. tenuis* was added. There were five replicates for each combination of density, sex, and age at the two experimental temperatures. After a 24 h period, the number of eggs consumed by the predator was scored. The consumed eggs were easily distinguishable, being either reduced to a hollow shell or completely absent.

### 2.3. Data Analysis

The mean number of prey consumed was plotted as a function of prey density, and the data were fitted to Rogers and Royamas’ random predator equation:(1)$$N_{e}=N_{0}\left(1-\exp\left[\alpha \left(T_{h}N_{e}-T\right)\right]\right)$$

*N_e_* is the number of prey/eggs consumed, *N*_0_ is the initial prey density, $T_{h}$ is the handling time, *T* is the duration of the experiment (24 h), and *α* is the attack rate.

To simplify the above equation, we used the “Lambert-W” function *W(x)* [50]:(2)$$N_{e}=N_{0}-\frac{W\left[\left(aT_{h}N_{0}\exp\left[\alpha \left(T_{h}N_{e}-T\right)\right]\right)\right]}{aT_{h}}$$

The values for the parameters *α* and *T_h_* were estimated through a non-linear fit of the data to the equation provided above in Python, accounting for the standard deviation of the data. The determination coefficient, R^2^, was calculated in addition to the parameters’ errors. A Tukey honest significance test (at a confidence level of 0.95) was performed to assess the significance of differences between different attack rates and handling times.

The statistical analysis concerning the effect of age, sex, and predator species on the prey consumption at each temperature was quantified through a three-way analysis of variance (ANOVA, type I, F-test statistic), accounting for a generalized linear model (GLM) with quasipoisson distribution. The analysis was performed in R (version 4.3.1; R core team 16 June 2023).

## 3. Results

### 3.1. Functional Response of M. pygmaeus on Eggs of T. absoluta

The functional response data of adult females and males of *M. pygmaeus* on eggs of *T. absoluta* was successfully fitted to Royama (1971) and Rogers (1972) equation and consequently exhibited a Type II functional response (Figure 1, Table 1). 

As shown in Table 1, the estimated values of the attack rate in 1-day-old females were 0.70 and 0.51, and those of the handling time were 0.31 and 0.27 h at 20° and 25 °C, respectively, whereas in males, the respective estimated values of the attack rate were lower (0.09 and 0.10) and those of the handling time were longer (0.57 and 0.33 h). 

In 10-day-old females, the estimated values of the attack rate were 0.33 and 0.36 and those of the handling time were 0.28 and 0.24 h, respectively, at 20 °C and 25 °C, whereas in males, the respective values of the attack rate were 0.19 and 0.08 and those of the handling time were 0.61 and 0.42 h, respectively.

Our results show that females exhibited higher attack rates and shorter handling times than their male counterparts at both tested temperatures and age groups. 

The lowest value of the attack rate of females was 0.33 and the highest was 0.70. The females’ shortest handling time was 0.24 h and the longest was 0.31 h. In comparison, the lowest attack rate value obtained for males was 0.08, and the highest attack rate, 0.19, was much lower than the values for females. The shortest handling time values for males were 0.33 h and the longest was 0.61 h, the latter being almost double the handling time of the females.

### 3.2. Functional Response of N. tenuis on Eggs of T. absoluta

As in *M. pygmaeus,* the functional response data of adult females and males of *N. tenuis* on eggs of *T. absoluta* was successfully fitted to the Royama (1971) and Rogers (1972) equation (Figure 2, Table 1). For *N. tenuis*, the estimated values of the functional response parameters of the attack rate (α) and handling time (*T_h_*) were lower than the respective ones for *M. pygmaeus* (Table 1). 

In 1-day-old females of *N. tenuis*, the estimated values of the attack rate (α) were 0.22 (compared to 0.70 for *M. pygmaeus (M.p.)*) and 0.14 (0.51 for *M.p.*) and those of the handling time were 0.18 (0.31 for *M.p.*) and 0.10 (0.27 for *M.p.*) hours at 20° and 25 °C, respectively, whereas in males, the respective estimated values of the attack rate were 0.20 (0.09 for *M.p.*) and 0.10 (0.10 for *M.p.*) and those of the handling time were 0.24 (0.57 for *M.p.*) and 0.19 (0.33 for *M.p.*) hours. 

In 10-day-old females, the estimated values of the attack rate were 0.18 (compared to 0.33 for *M. pygmaeus (M.p.)*) and 0.09 (0.36 for *M.p.*) and those of the handling time were 0.25 (0.28 for *M.p.*) and 0.14 (0.24 for *M.p.*) hours, respectively, at 20° and 25 °C, whereas in males, the respective values of the attack rate were 0.09 (0.19 for *M.p.*) and 0.04 (0.08 for *M.p.*) and those of the handling time 0.37 (0.61 for *M.p.*) and 0.19 (0.42 for *M.p.*) hours.

Our results show that in *N. tenuis*, as in *M. pygmaeus*, females exhibited higher attack rates and shorter handling times than their male counterparts at both the tested temperatures and age groups.

### 3.3. Comparison of Predation Efficacy of M. pygmaeus and N. tenuis

Three-way ANOVA indicated that the effects of predator species, sex, and age at both the experimental temperatures and the interactions of predator * sex and predator * age at 20 °C on the consumption of *T. absoluta* eggs were significant (Table 2). 

## 4. Discussion

Functional response experiments illustrate the relationship between the initial prey density and the number of prey killed or parasitized. Based on this, the type of functional response with its parameters, the attack rate and the handling time, are estimated. From this determination, valuable practical conclusions can be derived about the suitability of a potential biological control agent against a pest of a crop. The dynamics of these predator–prey interactions are defined by a plethora of factors, such as predator age and sex, developmental instar, and prey species [49]. In our experiments, we determined the functional responses and estimated the values of the attack rate and handling time of newly emerged (1 day old) and old (10 days old) *M. pygmaeus* and *N. tenuis* adult females and males feeding on different densities of *T. absoluta* eggs laid on tomato leaves. We found that, in *M. pygmaeus*, the estimated values of the attack rate were higher and those of the handling time were lower than the respective values for *N. tenuis.* The age and sex of both predator species affected the functional response parameters, i.e., the handling time was lower and the attack rate was higher in females and young individuals than in males and old individuals. 

The functional responses of *M. pygmaeus* and *N. tenuis* have been studied by other researchers following different experimental designs, prey densities, and host plants [39,40,41,43,44,45,46,47,48]. Our findings are in accordance with the functional response type obtained in a study by Sharifian et al., 2015 [40], where 1-day-old adults of unspecified sex of *M. pygmaeus* and *N. tenuis* fed on different densities of *T. absoluta* eggs at 25 °C. However, in this study, the values of the attack rate and handling time did not differ between *M. pygmaeus* and *N. tenuis* and were quite different from those obtained in our study. These observed differences between the obtained results from our study and the aforementioned study may be due to, among other factors, the different designs of the experimental arenas and the different sexes of the experimental predators. It is known that the arena size and the varying spatial distribution of the prey affect functional response parameters, namely the attack rate [42,51,52]. The unique experimental design described in our experiments was adopted in an attempt to align the experimental conditions more closely with field conditions, where the predator consumes *T. absoluta* eggs laid by adult females freely on both sides of the leaf surface. In contrast, Sharifian et al. (2015) utilized an experimental arena consisting of three tomato leaflets in a Petri dish, onto which the desired densities of eggs were added. These factors may provide possible explanations for the differential values of the functional response parameters.

In our functional response experiments, the predators *M. pygmaeus* and *N. tenuis* preyed on *T. absoluta* eggs and showed a Type II response. Prey type and even prey developmental stage may modify the functional response of predators. *M. pygmaeus* adult females have been shown to exhibit a Type II functional response when feeding on 2nd–4th instar nymphs of *Myzus persicae* Sulzer (Hemiptera: Aphididae) [43,47], pupae of *T. vaporariorum* [44], eggs of *E. kuenhiella*, and eggs of *T. absoluta* [41], the latter of which we find our results to be in accordance with; and a Type III functional response when feeding on eggs of *T. vaporariorum* [45]. The adult females of *N. tenuis* have been shown to exhibit a Type II response when feeding on *T. vaporariorum* nymphs [53], *Tetranychus urticae* Koch (Tetranychidae) adults [45], and eggs of *T. absoluta* [40], the latter in accordance with the results of our experiments; and a Type III response when consuming eggs and nymphs of *Bemisia tabaci* Gennadius (Homoptera: Aleyrodidae) [48]. 

In certain predator insects and mite species, the functional response is affected by adult age. The functional response of the predator mite *Amblyseius swirski* Athias-Henriot (Acari: Phytoseiidae) feeding on *T. urticae* was of Type II for a major part of its life, with the exception of 12-day-old females that exhibited a Type III functional response. This was attributed to the greater energy demands due to the higher reproductive output of females at that age [46]. A Type III response was also obtained with 1-day-old mated females of the parasitoid *Trichogramma brassicae* Bezdenko (Hymenoptera: Trichogrammatidae) when parasitizing *E. kuehniella* eggs, and a Type II response with 2–9-day-old females, which may be due to the effect of age progression on the searching efficacy (varied effect) and handling time (increase) [54]. For the mirid bugs, *Dicyphus bolivari* Lindberg and *Dicyphus errans* Wolff (Hemiptera: Miridae), both fifth-instar nymphs and seven-day-old females feeding on *T. absoluta* eggs exhibited a Type II response [55]. In the case of *M. pygmaeus*, the last nymphal instar, and adult females, have a general tendency to consume greater numbers of *M. persicae* Sulzer [56]. In our experiments, *M. pymaeus* and *N. tenuis* females and males maintained a Type II functional response as newly emerged (1 day old) and old (10 day old) adults.

We determined the functional responses of *M. pygmaeus* and *N. tenuis*, at two temperatures (20 °C and 25 °C) and found a Type II functional response with no substantial differences in the estimated values of the attack rate and handling time between the two temperatures. It has been noted that changes in the ecological settings may be reflected in the functional response and associated parameters [51]. Investigations into the temperature thresholds of dicyphine species have determined that those vulnerable to low temperatures are more resilient to high temperatures and vice versa. The mirid, *N. tenuis*, is considered the most thermophilous of the dicyphine species that have been investigated thus far, exhibiting optimum performance at temperatures between 20 and 30 °C [57], with individuals reaching adulthood at a maximum temperature of 35 °C [5]. Interestingly, in the context of the functional response, the attack rate and handling time are expected to be optimized at intermediate temperatures [58,59]. As such, *N. tenuis* females have been indicated to exhibit a Type II response when consuming *B. tabaci* (Hemiptera: Aleyrodidae) at relatively low temperatures (15 °C and 25 °C) and a Type III response at a higher temperature of 35 °C [60]. Conversely, *M. pygmaeus* maintained a Type II functional response over a range of temperatures [5]. 

The analysis of our results shows that in both predatory species, sex and age have a significant effect on the consumption of eggs at 20 °C. However, this significant interaction was not noted at 25 °C.

We provide the first evidence of a *M. pygmaeus* population outperforming a population of *N. tenuis*. The previous history of an individual, such as the previous host or prey that it was reared on, may influence its behaviour during the functional response experiment [42]. The two species we examined were reared in the laboratory on bean pods and *E. kuehniella* eggs. A preparatory period, where the predators were allowed to remain on tomato plants and feed on *T. absoluta* eggs, could have benefited and altered the outcome of the experiments.

Our findings indicate that both young and old males and females of the two predatory species exhibit a Type II functional response at the two experimental temperatures. However, the two species have different functional response characteristics to their prey, mainly depending on their sex and age, which may influence their role as biological control agents. 

In conclusion, our results provide evidence of the combined effect of sex and ageing on the functional response parameters of *M. pygmaeus* and *N. tenuis* at two temperatures and show that in both predator species, females have higher predation efficiency than males. In addition, newly emerged females have higher predation efficiency than 10-day-old females. Based on the estimated values of the attack rate and handling time, we conclude that *M. pygmaeus* may have a better predation efficiency than *N. tenuis* for eggs of *T. absoluta* on tomato. However, further field experiments are required in order to verify our present laboratory results. 

## Figures and Tables

**Figure 1 insects-15-00485-f001:**
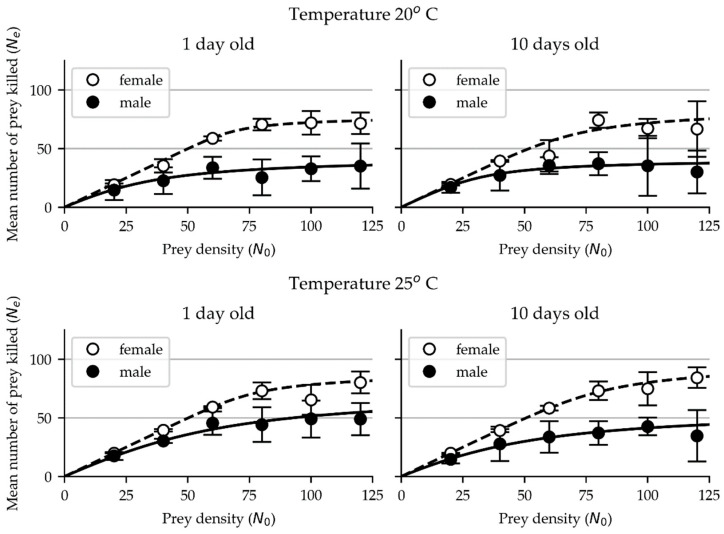
Functional response of 1-day-old and 10-day-old females (open circles) and males (closed circles) of *M. pygmaeus* on eggs of *T. absoluta* at 20 °C and 25 °C. Error bars represent standard deviation (SD).

**Figure 2 insects-15-00485-f002:**
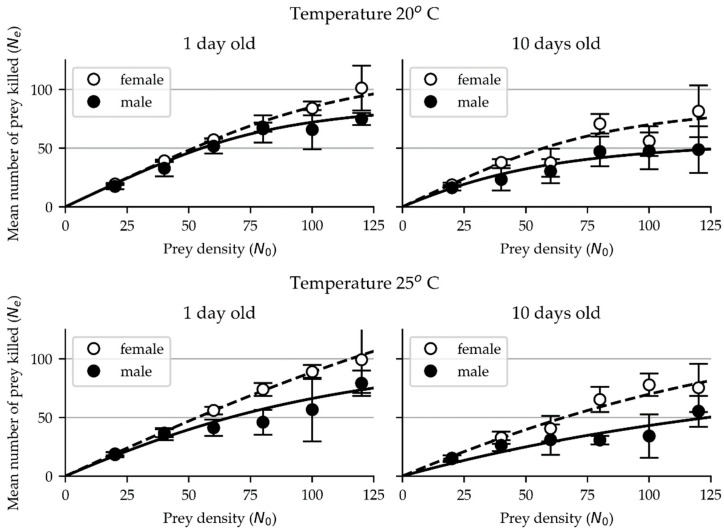
Functional response of 1-day-old and 10-day-old females (open circles) and males (closed circles) of *N. tenuis* on eggs of *T. absoluta* at 20 °C and 25 °C. Error bars represent standard deviation (SD).

**Table 1 insects-15-00485-t001:** Effect of sex, adult age, and temperature on the attack rate (α) and handling time (*T_h_*) of *M. pygmaeus* and *N. tenuis* adults feeding on eggs of *T. absoluta* on tomato leaflets. At each column, values followed by different letters are significantly different at *p* < 0.05.

Species	Sex	Age (Days)	Temperature (°C)	*α* (±SE) (h^−1^)	*T_h_* (±SE) (h)	R^2^
*M. pygmaeus*	female	1	20	0.70 ± 0.26 a	0.31 ± 0.02 a	0.99
	female	1	25	0.51 ± 0.12 b	0.27 ± 0.02 a	0.83
	female	10	20	0.33 ± 0.09 c	0.28 ± 0.03 a	0.92
	female	10	25	0.36 ± 0.04 c	0.24 ± 0.01 a	0.99
	male	1	20	0.09 ± 0.04 d	0.57 ± 0.08 b	0.81
	male	1	25	0.10 ± 0.01 d	0.33 ± 0.03 a	0.93
	male	10	20	0.19 ± 0.08 e	0.61 ± 0.05 c	0.71
	male	10	25	0.08 ± 0.01 f	0.42 ± 0.03 d	0.88
*N. tenuis*	female	1	20	0.22 ± 0.02 g	0.18 ± 0.02 e	0.99
	female	1	25	0.14 ± 0.01 g	0.10 ± 0.01 f	0.99
	female	10	20	0.18 ± 0.07 g	0.25 ± 0.06 a	0.87
	female	10	25	0.09 ± 0.01 d	0.14 ± 0.03 a	0.96
	male	1	20	0.20 ± 0.07 g	0.24 ± 0.03 a	0.98
	male	1	25	0.10 ± 0.03 d	0.19 ± 0.06 e	0.88
	male	10	20	0.09 ± 0.01 d	0.37 ± 0.05 g	0.93
	male	10	25	0.04 ± 0.01 d	0.19 ± 0.11 e	0.88

**Table 2 insects-15-00485-t002:** Three-way ANOVA (quasi-poisson distribution) on factors affecting consumption of eggs of *T. absoluta* by *M. pygmaeus* and *N. tenuis* at 20 °C and 25 °C.

Source	df	Deviance	*F*-Value
20 °C			
Predator	1	211.73	21.27 a
Sex	1	591.80	59.44 a
Age	1	104.83	10.53 b
Predator * Sex	1	88.28	8.87 b
Predator * Age	1	51.49	5.17 c
Sex * Age	1	5.82	0.58
Predator * Sex * Age	1	11.89	1.19
25 °C			
Predator	1	52.80	3.84 c
Sex	1	578.81	42.19 a
Age	1	136.81	9.97 b
Predator * Sex	1	24.74	1.80
Predator * Age	1	24.66	1.79
Sex * Age	1	23.53	1.72
Predator * Sex * Age	1	5.70	0.42

a: *p* < 0.001, b: *p* < 0.01, and c: *p* < 0.05.

## Data Availability

Data is unavailable due to privacy restrictions.

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
