# Peer review of "Effect of Sex, Age and Temperature on the Functional Response of Macrolophus pygmaeus Ramber and Nesidiocoris tenuis Reuter (Heteroptera: Miridae) on Eggs of Tuta absoluta"

_insects, 2024, doi:10.3390/insects15070485_

Round 1

Reviewer 1 Report

Comments and Suggestions for Authors

This short but informative manuscript represents the results of the experimental study on functional responses of two predatory bugs (Macrolophus pygmaeus and Nesidiocoris tenuis) preying on Tuta absoluta eggs in relation to temperature, adult age, and sex. The experiments were well designed and conducted. The statistical analysis is correct (although not sufficient). The text is well and clearly written. Although the functional responses of these predators have been studied by other researchers, the authors obtained new important data. The results of the study can be used for the planning of the use of these biocontrol agents against Tuta absoluta in greenhouses. Thus, the manuscript can be published although it needs some minor corrections and improvements (see below).

Lines 102-106: Consider replacing this text to the Discussion, as it concerns not reasons and aims (as should be in the Introduction) but results of the study.

Line 124: “a large number of” can be deleted: “approximately 300 T. absoluta  adults” is enough.

Line 133: “prey freely navigate” sounds strange when applied to eggs.

Lines 171, 172, 177, 178 and below in the Results section:  Pairwise comparisons such as “lower”, “longer”, “higher” “shorter” etc. should be supported by statistical significance either in the text or in corresponding tables or figures. ANOVA of the total data set (Table 2) is not enough to support all particular pairwise differences.

Lines 187-188: Please, explain in the legends what kind of descriptive statistics is used: mean and SEM or mean and SD?

Table 1: Please, evaluate statistical significance of pairwise differences between the data in each column (by the Tukey’HSD or some other appropriate test) and give this information in the table in a usual way (values in the same column labeled with different letters are significantly different at p<0.05 level).

Lines 213-214: Please, consider replacing this sentence to the Discussion because (as far as I understand) it is not directly based on the experimental results of the present study. Predator efficiency can be only evaluated by corresponding field (or greenhouse) experiments. In this context, I do fully agree with the final statement in lines 318-319.

Lines 217-218: The same question: please, explain in the legends what kind of descriptive statistics is used: mean and SEM or mean and SD?

Line 280: As Trichogramma wasps are not predators but parasitoids, “feeding” should be replaced by “parasitization”.

Reviewer 2 Report

Comments and Suggestions for Authors

Major comments:

It is not clear how this work is novel compared to previous studies in which the functional response was also assessed.

Authors states that the primary objective of this study was to access the predatory efficacy of the two species, M. pygmaeus and N. tenuis (lines 98-99). However, the concept of effectiveness in IPM goes far beyond what this study offers. It would be important to understand the reason why such an objective is formulated in this way. I suggest rewriting the objective.

More details about the stock population must be provided, e.g. Where the individuals were collected, how long they were kept in the laboratory conditions before the experiments began and why they were not fed with T. absoluta eggs, supplemented with tomato plants.

Authors states that preliminary experiments were carried out to access voracity of the predators (lines 136-137). Authors must mention what results they obtained and to what extent they were important for their experimental design. I ask if it wouldn't make sense, instead of these experimental tests, to look in the literature for maximum daily values in terms of number of eggs consumed (E.g. DOI:10.1007/s12600-024-01130-0 and DOI:10.1002/ps.7635).

It is stated that functional response experiments were conduced at seven predefined densities of T. absoluta. Authors should clarify: i) the egg densities offered in each treatment and ii) why only 6 densities are presented on the graphs 1 and 2.

Statistical analysis is missing over the result section: The significance of voracity under different prey densities (from what density of T. absoluta eggs, voracity reach the maximum), ii) significance of the regression model (for Type II functional response) and the variance explained by the model expressed by the coefficient of determination and iii) how significant are the differences between the various search rates and handling times, maybe by using the Lower and Upper values of 95% Confidence Intervals.

Lines 305-306: What result obtained in this study allows us to affirm this?

Additional comments:

Lines 99-106: I consider this part of the text out of context. I would recommend deleting it.

The section 2.2 on Material and Methods is confusing and needs to be rewritten. For instance, the first paragraph should be divided into 2, once there is one section describing how eggs of T. absoluta were obtained and the second part describe the bioassays itself. Furthermore, it is only in the 2nd paragraph that we realize the characteristics of the experimental arenas used in the experiments.

Lines 195-214: this section intended to analyze predation by N. tenuis on eggs of T. absoluta. However, it is not clear why the results obtained are contrasted with those obtained for M. pygmaeus. I believe that this comparative discussion should be part of the Discussion section.

The texts presented on the headlines 3.1 and 3.2 (lines 165 and 194) are misleading. In fact, the authors along these 2 sections analyze the searching rate and handling time, not the predation values obtained.

The authors should include more current references. I provided 2 examples. On lines 69-71 authors cite Desneux et al., 2010. However, a more recent paper provide more updated information (e.g. Desneux N, Han P, Mansour R, Arno J, Brevault T, Campos MR et al., Integrated Pest Management of Tuta absoluta: practical implementations across different world regions. J Pest Sci 95: 17–39 (2022).

The manuscript must be revised to remove little problems. I will give a couple of examples: line 53, do not start the sentence with M. pygmaeus, line 70 North Africa instead N.Africa, Line 133, replace the word navigate, 177, the word markedly should be replaced or erased.

Reviewer 3 Report

Comments and Suggestions for Authors

This is a very good study, well executed and well presented.  However, there are some clarifications that, although not difficult,  are important before the paper is published.

The Introduction is good and gives a preview of conclusions, which is good style.

line 118.  how were assay temperatures chosen? Explain.

l 128. give vial dimensions

l 145. give n values.  how many replicate cups (experimental units) were used for each factor level combination?  

l 144-160. was density a factor?  Clarfiy and If so, give levels and how they were chosen. 

Figure 1. I suggest adding a visual key to open and closed circles inside the graph itself, not in the figure legend.  Specify the error bars- they should ideally represent 95% c.i. of the mean using the SE.

Table 2.  this is not a standard ANOVA table and should be corrected to show error terms used in calculating F values and show P values.  What is "deviance"??

l. 222 Describe the nature of the significant interactions here and their implications in discussion

Reviewer 4 Report

Comments and Suggestions for Authors

Dear Authors,

Your submitted material is very interesting and could represent a contribution to the science. The text is written in an understandable English language.

However, I could conclude that your work will be worth of publishing only if some sections are significantly improved. The design of experiment is not clearly explained and there are issues that has to be addressed before this material is accepted. Please find below my comments:

Abstract and simple abstract: Please keep text consistent. Correct (Lep.: Gelechiidae) to (Lepidoptera: Gelechiidae) and add whiteflies the order and the family. Replace English name with its Latin name.

It should be clarified in abstract what is Type II or if it is too complicated, it should be deleted from this part of manuscript. Please do not start sentence with abbreviation such as abbreviated names of species. In Simple abstract the species are abbreviated when they were mentioned for the first time. Please correct.

Keywords: Please revise you list of words. Words tomato, sex, temperature etc. are not specifying to this topic. Also, full names should be used.

L37 Do you think they are native or autochthonous species? Please check if the word endemic is adequate for this context.

L38 Please do not start sentence with abbreviation. Please act accordingly in the whole manuscript.

L41 Missing space between bracket: e)[2

L45 This regime could not be considered as omnivorous. Please replace with an adequate term.

L51 I am not very convinced that this ability is advantage. It could be considered as advantage only for the insect but for agricultural production is disadvantage.

L56 Coma is missing after regions.

L57 and L57 The families of the plants should be uniformly presented. Please revise the whole manuscript and format it.

L65 Please give both Latin or both in English.

L67 Please correct microlepidodpteran.

L70 Missing space after N.

L85-87 It is confusing only mentioning these two categories. Please give more explanation.

L97 Missing space after coma.

L99 You are not fighting against eggs but against the species suppressing it in the egg stage. Please correct this.

L100 Missing space- T.absoluta

L110 Name and location of laboratory?

L102-106 These are results and do not belong to this section. Please delete.

Material and method: this section suffers of lack of data and explanation. It is not clear how the whole experiment was designed. It is not acceptable to say that approximately 300 insects were used. This should be given precisely. If the authors do not know precisely than need to be explained why did you give only approximation? There is not given number of replicates. How many males? How many females? How did the others recognize parasitized eggs? How did you count? The material and method requires to be significantly improved.

L153 Please revise the context.

L162 Please delete relevant.

Figure 1 should be improved. It is blurry. What is open and what is closed? Better would be to say black and white.

In the whole section of Results: Please explain why did you give approximation of results (~). That should be replaced with the precise results.

Table 1. There is enough space in table to give full words males and females.

Subsection 3.2. Comparison of these two species should not be given in this part because it is very hard to follow the text. Please first give the results about the N. t. and then give the comparison.

L207 Missing space before bracket.

L220-L223 This sentence is very confusing. It requires simplified explanation.

Table 3. This table needs to be formatted. Numbers under df escaped on another side. What do the superscript letters mean? Why don’t you have these letters in all parameters? It should be more clarified why so that the readers understand what the authors wanted to present here. There is a lack of other statistical parameters in this analysis, such as SE etc.

L251 Which other factors? If you mention them, please specify them.

L257 Coma is not needed here - et al.,

L260 There is not accepted correction from the track changes.

L262 type II or Type II? Please choose one of those and correct it in the whole manuscript.

L266-267 This is not clear. Please modify the sentence. Also L270. From the language accuracy point of view, it is not accurate to say that results obtained much before your study, are in accordance with it. Your study is in accordance with the previously published results.

L316 Please replace once with females. 

Comments on the Quality of English Language

English is understandable. 

Round 2

Reviewer 2 Report

Comments and Suggestions for Authors

I found a significant improvement in the manuscript. However, I have too comments:

1.     Authors state: “From the literature review, it appears that to date, similar studies determining the functional responses do not refer to the sex and age of the predatory insects”. This is not true. Let me give two examples for ladybirds and one for Nesidiocoris tenuis:

Cabral, S., A.O. Soares & P. Garcia (2009). Predation by Coccinella undecimpunctata L. (Coleoptera: Coccinellidae) on Myzus persicae Sulzer (Homoptera: Aphididae): effect of prey density. Biological Control, 50(1): 25-29. doi: 10.1016/j.biocontrol.2009.01.020.

Moura, R., P. Garcia & A.O. Soares (2006). Does pirimicarb affect the voracity of the euriphagous predator, Coccinella undecimpunctata L. (Coleoptera: Coccinellidae)? Biological Control, 38: 363-368. doi: 10.1016/j.biocontrol.2006.04.010.

Gavkare, O., Sharma, P. L., Sanchez, J. A., & Shah, M. A. (2017). Functional response of Nesidiocoris tenuis (Hemiptera: Miridae) to the two-spotted spider mite, Tetranychus urticae. Biocontrol Science and Technology, 27(9), 1118–1122. https://doi.org/10.1080/09583157.2017.1385054

My previous comment on the lack of originality of the study stands and for, that reason, I recommend to authors to be more carefully on that.

2.     The authors have contextualized their study in a more applied context. Which is a suitable strategy. In this context, and once the authors have calculated prey consumption of predators on T. absoluta eggs, I recommend mentioning previous studies in which this were already calculated (E.g., for M. pygmaues: DOI:10.1007/s12600-024-01130-0 and DOI:10.1002/ps.7635, https://doi.org/10.1111/j.1439-0418.2008.01319.x¸ https://doi.org/10.1007/s10340-013-0498-6).

Reviewer 3 Report

Comments and Suggestions for Authors

Everything looks good now, except for the error bars in the figures.  The figure legend does not specify what the error bars mean.  In their response, the authors say the bars are SD, but this is not stated in the manuscript.  In addition, SD is not a good choice for error bars, which are more meaningful if they represent a 95% CI of the mean, based on the SE.

Reviewer 4 Report

Comments and Suggestions for Authors

The manuscript is fine.

Comments on the Quality of English Language

No additional comments

Author Response

Reviewer 4 has no additional comments